# Visualizing the Knowledge Domain in Health Education: A Scientometric Analysis Based on CiteSpace

**DOI:** 10.3390/ijerph19116440

**Published:** 2022-05-25

**Authors:** Boyuan Chen, Sohee Shin, Ming Wu, Zhihui Liu

**Affiliations:** 1Department of Physical Education, Henan University of Science and Technology, Luoyang 471023, China; cby@haust.edu.cn; 2School of Sport and Exercise Science, University of Ulsan, 93 Daehak-ro, Nam-gu, Ulsan 44610, Korea; 3School of Physical Education (Main Campus), Zhengzhou University, Zhengzhou 450001, China; wumingzzu@126.com (M.W.); zhihuiliuzzu@163.com (Z.L.)

**Keywords:** health education, CiteSpace, scientometric, health portion, COVID-19

## Abstract

Objectives: This study aimed to visualize the evidence in the global research on health education to better improve the nation’s health literacy and to guide future research. Method: We searched the Web of Science (Core Collection) electronic databases. The search strategies: topic: (“Health Education” OR “Education, Health” OR “Community Health Education” OR “Education, Community Health” OR “Health Education, Community”) AND document: (Article) AND language:(English). Articles of evidence from January 2011 to December 2021 with those words in the title or abstract or keywords will be included in this review. We used the Citespace 5.6.R5 (64-bit) to investigate and determine the thematic patterns, and emerging trends of the knowledge domain, and presented a narrative account of the findings. Result: We analyzed 10,273 eligible articles. It showed that BMC Public Health displays the most prolific journals. Author MARCO PAHOR is highlighted in health education. The University of Sydney has published the most studies about health education. The USA plays an important role in these studies. Specifically, the visualization shows several hotspots: disease prevalence surveys and a specific population of knowledge, attitude and practice surveys, health intervention, chronic and non-communicable management, youth-health action, sexual and reproductive health, and physical activity promotion. Furthermore, document co-citation analysis indicated that there are 10 main clusters, which means the research front in health education. Meanwhile, by the citation detected, COVID-19, has achieved universal health coverage in related studies, however, public health education and the health workforce might be more popular in the coming years. Conclusion: Health education is an effective measure to shift the concept of public health and improve healthy living standards. The present study facilitates an extensive understanding of the basic knowledge and research frontiers that are pivotal for the developmental process of health education and allows scholars to visualize the identification modes and tendencies.

## 1. Introduction

The World Health Organization (WHO) guideline [1] stated: “A prime purpose of health education (HE) is to assist people in improving health status by acquiring more information, affecting motive, and uplifting health literacy (HL).” It means that the purpose of HE is to solve our own health problems effectively. From an individual’s perspective, HE can help people acquire new modern health concepts, develop positive and healthy behaviors, motivate them to decide on a healthy lifestyle, enhance self-protection awareness, promote communicable disease prevention, and support chronic non-communicable disease management. From a social perspective, HE is an effective measure for alleviating the strain on medical. It helps people to acquire health-related knowledge and skills, meet people’s growing demand for mental health services, and assists in curbing increasing medical costs.

However, we must be faced with the truth that in most schools, HE-related areas of knowledge account collectively for little more than 5% of children’s total school experience. Over-practical or over-preventative measures seem to have done little in enhancing the quality of the public’s understanding or their engagement with practitioners [2]. Moreover, a lack of training for both trainers and health professionals in the educational aspects of health improvement schemes has led to many schools having too few staff who are competent in this area [3]. Although the Health People 2020 prospect and Sustainable Development Goals 2030 have raised new hopes for a world free of poverty and poor health, the HE problem is not new, however, it remains to be acted on. To reverse this unfavorable situation, levels of HL need to be raised throughout the world, especially in developing countries. With the COVID-19 effects on societies and economies, we have witnessed the greatest global health challenge in a generation [4]. Many countries have had the best responses to the COVID-19 pandemic and achieved good goals. However, HE, especially healthcare management education is nascent or non-existent in many countries, which lead to some failures concerning COVID-19 to some extent, cities were near paralyzed, and citizens were disappointed. Unfortunately, this has been further exacerbated by serious errors in practice, and significantly, spurious media reports, which made it more and more difficult to persuade people to alter their lifestyle or avoid known trigger factors for the disease. It could be proved that improvements in the standards of HL over the past decade have been unsatisfactory [2]. Studies of scientific literature demonstrate a well-established correlation between HL and HE [5] so HE has received widespread attention as a focus of research again. It provides us an opportunity to go back and grasp its research dynamics, think about the roles of public health and HE, and lay the foundations for subsequent innovative research. Although some scholars have reviewed and evaluated the potential benefits of HE from different perspectives, most of them are based on a qualitative review of direct control of the literature, and there is a large degree of subjectivity in the selection and classification criteria, and hotspots.

Therefore, this study analyzes the existing HE research based on the Web of Science (Core Collection) database. The primary purpose is to map the knowledge domain of HE research, investigate collaborations related to HE, reveal thematic patterns and hotspots, and explore the research frontiers. Meanwhile, to assist professionals, we also hope to quantify the pathways to find new trends in HE research and help scholars better tackle health service challenges for a ‘well-being society’. All the glossary of terms in this study was shown on Table 1.

## 2. Bibliographic Records and Method

### 2.1. Data Source Collection

In this study, the data was downloaded on 5 January 2022, from the Web of Science (Core Collection). The MeSH and entry terms were used singularly or in combination through the following strategies: topic: (“Health Education” OR “Education, Health” OR “Community Health Education” OR “Education, Community Health” OR “Health Education, Community”) and their Boolean combination, which means that articles with those words in the title or abstract, or keywords will be retrieved. Refined by document: (Article) AND language: (English). The data set was collected for Timespan = 2011–2021. From the search, we included 10,273 records using the strategy settings. 

Table 2 lists the search queries and strategies procedure. Each downloaded article was saved as plain text files for subsequent data processing and analysis and transformed into an executable format using the “Data/Import/Export” of Citespace for visualization.

### 2.2. Software Program and Method

Citespace, designed by Chaomei Chen, was utilized to analyze and visualize the theme architectures and investigate hotspots of the selected articles. The primary source of input data for this software was the Web of Science database, and the primary procedure steps of this program were time slicing, thresholding, model establishment, pruning, merging, and mapping [6].

The software could determine research frontiers and forecast bursts by titles, abstracts, descriptors, and bibliographies, utilizing three core terms: co-occurring keywords, document co-citation clusters, and citation bursts. Those terms could address three challenges timely and practically: marking keywords, determining the essence of study frontiers, and determining new tendencies and abrupt variations [7]. Hence, the process made it simple for researchers to identify hotspots by recognizing nodal points with high frequency and betweenness centrality (BC). When the clustering function was initiated, it showed the frontiers of the research, the Modularity Q, and Mean Silhouette scoring which remarkably influenced visualization, reflecting the general structure feature of the net. Normally, Q > 0.3 denotes a generally significant structure. If S ≥ 0.5, the cluster will be deemed reasonable. Moreover, emerging trends could be tagged by using a citation burst [8].

The present study mainly compared and analyzed the characteristics of the data concerning 10,273 HE-related articles in the Web of Science (Core Collection) using Citespace 5.6.R5 (64-bit). We first identified author/institution/country collaborations, then examined co-occurring keywords, then explored major clusters with document co-citations in HE, and then detected emerging trends through reference burst citations. Studies published from 2011 to 2021 set the parameters as chosen with a time slice of 1 year for the analysis, and the selection criteria were the top *n* = 50 per slice.

## 3. Results

### 3.1. Publication Years and Journals

Since 2011, the SCI database has included 10,273 HE articles and manuscripts, and increasing attention has been paid to the accumulative published articles about HE, see Figure 1.

The ten prolific journals with the greatest number of published articles about HE studies are listed in Figure 2, which can provide new researchers with more important information when they consider a submission. Journals of BMC PUBLIC HEALTH ranked first in terms of the quantity of published research. These journals aim to propagate epidemiological knowledge and all-around information pertaining to public health. They especially highlight the societal determining factors of health, and the environment-, behavior-, and occupation-related associations with illnesses, as well as the influence of healthcare strategy, practice, and intervention on communities, whereas there is not any clinical research in BMC Public Health journals. The INTERNATIONAL JOURNAL OF ENVIRONMENTAL RESEARCH AND PUBLIC HEALTH (IJERPH) ranked No. 3. As an all-around multidisciplinary periodical, IJERPH had a 2020 impact factor of 3.390 according to the Journal Citation Reports 2020 edition, which ranks 68/203 in ‘Public, Environment and Occupation Health’. IJERPH highlights the research on the interplay between environmental health and life quality, and the socio-cultural, political, economic, and law aspects associated with environmental management, and public health.

### 3.2. Cooperative Network Visualization

Figure 3 can provide highly personalized scientific research information for other researchers. It is immediately apparent that many authors tend to collaborate with a relatively stable group of collaborators to generate several major author clusters, and each cluster usually contains two or more core authors. This analysis demonstrates that the most representative author in the field of HE is Marco Pahor, followed by Abby C King, Roger A Fielding, Anne B Newman, and Todd M Manini. Marco Pahor, M.D., is an internationally recognized geriatrician and epidemiologist expert on population-based studies, clinical trials, and multidisciplinary translational research in the fields of aging, disability, and cardiovascular disease. His h-index is 96 and his average number of citations per article is 89.87.

The University of Sydney has published the largest number of studies (162). The USA created the largest number of studies (3085). South Africa (1.15) has the most centrality in the network.

### 3.3. Co-Occurring Keywords Analysis

Co-occurring keywords reflect research hotspots in the field of HE. There are 103 individual nodes and 114 links in Figure 4. As we can see, the maximum frequency was ‘health education’ with 1878, followed by ‘prevalence’, ‘knowledge’, ‘health’, and ‘intervention’. Most nodes marked with purple circles represent good BC, and these keywords are also important, such as ‘awareness’ with a maximum centrality of 1.12, followed by ‘physical activity’, ‘reproductive health’, ‘knowledge’, and ‘adolescent’.

### 3.4. Document Co-Citation Analysis

The results are illustrated in Figure 5 (right). These nodes and lines represent cited articles and co-citation relationships among the whole data set, respectively. The more cited, the larger node. The color and thickness of the circle in the node indicate the citation frequency at different time periods. Line colors correspond directly to the time slice, meaning that cold colors represent earlier years, while warm colors show more recent years. For example, blue lines represent studies that were co-cited earlier. Recently cited studies are shown by yellow or orange lines. The citation year ring represents the citation history of this study; the color of the citation ring represents the color responding to the citation time, with a similar color code as explained before. The thickness of an annual ring is proportional to the number of citations in a time zone.

Table 3 presents the top five most cited articles in the field of HE. These cited articles mainly involved physical activity, global health, COVID-19, violence prevention, HL, and methodology. Firstly, the most cited paper tested the effectiveness of a long-term structured physical activity program, by Pahor M (2014) [9], which summarized a successful HE initiative for aging persons, involving a structured moderate-intensity physical activity program that showed positive aspects and reduced older adults’ major motor disabilities over 2.6 years. The second most cited paper was by Bray F (2018) [45], which provided a status report on the burden of cancer worldwide, identifying lung cancer as the first condition for both sexes to be careful about, with lung, prostate, colorectal, liver, and stomach cancers reported as high-incidence cancers among males, and breast, colorectal, lung, and cervical cancers for females. However, the main causes of cancer and the leading cause of cancer deaths varied considerably across different countries and within countries. Importance was attached to registering high-quality cancer data, particularly in low- and middle-income countries. The global status report on violence prevention 2014 [10] analyzed 133 countries’ data to determine national efforts to deal with interpersonal violence, such as child abuse, adolescent violence, intimate partner and sexual violence, and elder maltreatment. Sorensen K (2012) [11] reviewed the existing definitions and models on HL and developed a combined definition and concept model based on medical and public health. This definition and concept model could be useful to enhance the intervention effectiveness and assess different viewpoints of HL in healthcare, preventive medicine, and health promotion. Zhong. BL (2020) [12] investigated Chinese people’s knowledge, attitudes, and practices (KAP) during the outbreak of COVID-19, indicating that Chinese people had a positive KAP, particularly women who have high socioeconomic status.

### 3.5. Clusters Interpretation

The analyzed document co-citation cluster represents the main research patterns in the knowledge domain of HE. There are 1116 individual nodes and 2044 links which contain 10 main clusters. The modularity Q was 0.8837 and the Mean Silhouette was 0.3505, which means this cluster has a reasonable and significant structure [8], see Figure 5 (left).

Table 4 summarizes the details of the 10 clusters named, in which the log-likelihood ratio is popular with researchers and always showed the best results associated with clustering [7]. According to the document co-citation cluster markers, it can be observed that scholars and experts are concerned about disease prevention and control, global health training, intervention, and management. Combined with highly cited references, the topics of global HE, disease intervention, and healthcare work are at the hotspots of current research. HL, chronic disease management, health promotion, social media-based HE, COVID-19, and neglected tropical diseases (NTDs) have been studied horizontally and deeply.

According to the narrative summary of CiteSpace, we found that Cluster #0 (soil-transmitted helminth infection), Cluster #4 (global health training), and Cluster #9 (community health worker), have the strongest citation bursts in dark dots, suggesting that these clusters constitute the major study efforts in the period 2011–2021.

Cluster #0 is labeled as soil-transmitted helminth infection (STH), with more attention being given to STH infections and prevalence, the intensity of infection, and associated risk factors. There are a total of 77 articles in this cluster. The five most cited articles include Bieri FA (2013), Pullan RL (2014), Utzinger J (2009), Bethony J (2006), and Strunz EC (2014). Bethony J (2006) [46] introduced STH infections, discussed the current prevalence and drug interventions, and pointed out the significant need to develop and measure new control tools. Utzinger J (2009) [47] described and advocated the effects of educational, biomedical, and project strategies, and geospatial tools on achieving sustainable STH control. Bieri FA (2013) [13] developed a HE intervention program to expand students’ knowledge of STH, improve their HL, and advise behavioral changes to reduce prevalence and disease burden. Pullan RL (2014) [14] analyzed epidemiological data from 118 countries to estimate the number of infections and disease burden of STH infections globally in 2010. They found that STH infections and years of healthy life lost (YLDs) due to disabilities occurred mostly in Asia. They calculated that there were about 438.9 million people infected with hookworm, 819.0 million with A. lumbricoides, and 464.6 million with T. trichiura, with most of the cases occurring in Asia. Strunz EC (2014) [48] conducted a meta-analysis and concluded that water, sanitation, and hygiene access and practices are generally associated with reduced odds of STH infection.

Cluster #4 is labeled as global health training, and mainly includes the investigation of training programs, innovative model development, and evaluations on improving health and achieving equity in health for all people worldwide. There is a total of 49 articles in this cluster. The five most cited articles are by Frenk J (2010), Koplan JP (2009), Battat R (2010), Crump JA (2010), and Drain PK (2007). Due to the rapid demographic and epidemiological transitions, health professional education has not kept up the pace with these challenges, leading to traditional practitioners being unable or having the insufficient professional capacity to answer the fresh health challenges [15]. Koplan JP (2009) [16] highlighted the importance of using a common definition of global health and displayed the logic, connection, and function beneath the definition of global health. Battat R (2010) [49] reviewed competencies and educational approaches for teaching global health. They indicated that medical education pays more attention to competencies about understanding the global burden of disease (GBD), primary care within diverse cultural settings, healthcare disparities between countries, migrant health, travel medicine, and skills to better adapt to different cultures, populations, and healthcare systems. Crump JA (2010) [50] designed a set of guidelines for a field-based global health training program to help organizations, trainees, and institutions address various challenges. Drain PK (2007) [51] reviewed evidence of the benefits of promoting more global health teaching and opportunities to medical students and provided medical schools with several steps to meet the growing global health needs of medical students.

Cluster #9 is labeled as the community health worker, which focuses on health promotion. There are 44 articles in this cluster. The three most cited articles are Bray F (2018), Ferlay J (2015), and Patton GC (2016). Bray F (2018) [45] ‘s paper was described and explained in Section 3.4. Ferlay J (2015) [17] briefly described that lung, breast, and colorectal cancers were the most diagnosed, and lung, liver, and stomach cancers were the three common causes of cancer death, respectively. Patton GC (2016) [52] called for immediate investment in adolescent health and wellbeing and discussed how to create opportunities to meaningfully engage and increase health and wellbeing resources to address the challenges.

Important cited articles of other clusters in this knowledge domain are worth mentioning. These cited articles are more concerned with chronic disease and cancer incidence rates. Chobanian AV (2003) [53] revealed the 7th report of the Joint National Committee on the prevention, detection, evaluation, and treatment of high blood pressure to provide an evidence-based approach to the prevention and management of hypertension. Wild S (2004) [54] indicated that the prevalence of diabetes mellitus for all age groups worldwide will become more serious, estimating 366 million (4.4%) in 2030. The increase in the proportion of people >65 years of age probably might be the reason for the demographic change in diabetes. Meanwhile, they found that men had a higher prevalence of diabetes than women, but there are more women with diabetes than men. Ogden CL (2014) [18] compared trends in obesity among US children between 2003 and 2012 and provided a detailed analysis of obesity trends among US adults. Jemal A (2011) [19] reported that more than half of all cancer cases and cancer deaths occurred in developing countries. Furthermore, medical and health conditions in developing countries did not meet local needs. Specifically, the overall cancer incidence in developing countries is half that of developed countries, but overall, cancer mortality rates are broadly similar, which might be due to a lack of access to timely and standardized care, or advanced conditions at the time of admission.

### 3.6. Turning-Point Articles in Terms of BC

The broader rings in Figure 5 are another important sign of node, and play a ‘connected’ role in the visualization network, which is considered the novel and interdisciplinary turning point [55]. Table 5 shows the top three papers in terms of BC. More detailed information about the centrality can be found in Appendix A.

Murray CJL (2012). Disability-adjusted life years (DALYs) for 291 diseases and injuries in 21 regions, 1990–2010: a systematic analysis for the Global Burden of Disease Study 2010 [56], connects Cluster #0 (soil-transmitted helminth infection) and Cluster #2 (taenia solium). The average formulation years of the two clusters are 2011 and 2013, indicating STH is one of the knowledge bases of taenia solium. Murray CJL (2012) indicated that the global disease burden kept on moving from communicable to non-communicable diseases (NCDs) and from premature mortality to years lived with disability. In sub-Saharan Africa, however, many communicable NTDs, maternal health, neonatal, and malnutrition remain the major causes of disease burden. Mental problems and behavioral disorders, and chronic diseases should be paid attention to, which would bring out new issues or challenges for health systems.

Sorensen K (2012). Health literacy and public health: a systematic review and integration of definitions and models [11] described in Section 3.4, connects Cluster #2 (taenia solium) and Cluster #3 (COVID-19 pandemic). Sorensen K (2012) expressed that it is very difficult to horizontally compare analysis research results in different countries, due to there being no common results about the definition of HL or its conceptual dimensions. They reviewed and then developed a comprehensive conceptual model, containing 12-conceptual dimensions, such as the knowledge, motivation, and competencies of accessing, understanding, appraising, and applying health-related information within healthcare, disease prevention, and health promotion settings, respectively.

Braun V (2006). Using thematic analysis in psychology [20], Braun V (2006) described a qualitative research method, outlined the disadvantages and advantages of thematic analysis, and recommended thematic analysis as a useful and flexible method for qualitative research in and beyond psychology.

### 3.7. Citation Bursts

Burst detection could be widely used to explore the research trends of a research field [7], and recent ongoing bursts can indirectly predict future trends [57]. This study also utilized this algorithm to output the recent citation burst for exploring the emerging trends of HE studies. We set the Minimum Duration = 2 years and viewed the result. There were 141 references with the strongest citation bursts, see Appendix B. The burst group with the end year of 2021 suggests that their citation burst will probably continue in the future. In total, 31 papers with the end year of 2021 were classified to represent different emerging trends in the future. The main future directions based on the recent citation burst are summarized in Table 6.

The increasing compelling trend is COVID-19. Huang CL (2020) [61] and Zhu N (2020) [63] collected and described data on these patients with suspected 2019-nCoV in Wuhan, China offering the epidemiological, clinical, laboratory characteristics and treatment, and clinical outcomes evidence of 2019-nCoV at the early stages of the epidemic. Bai Y (2020) [69] analyzed a case of asymptomatic COVID-19 transmission from human-to-human infection. More importantly, they provided and highlighted the healthcare department’s need to pay attention to the potential effects of COVID-19, and suggested several strategies that can prevent, control, and intercept the spread of 2019-nCoV. Sohrabi C (2020) [70] reviewed the existing COVID-19 prevention, diagnosis, treatment, and prognosis knowledge, and suggested strict surveillance and ongoing monitoring to accurately track and potentially predict its future. Meanwhile, the surveys showed that most of the residents had better KAP towards COVID-19 [58,59,60], but those with comorbid conditions [65], i.e., elderly, lower-income, and other social gradients [62] were significantly lacking the necessary knowledge about COVID-19.

The second direction is to ensure healthy lives and promote well-being for all ages. The objective evaluation of disease burden could help the policymakers to determine the health problems which need to be solved and prioritize development projects which are based on the premise of fair, reasonable, and effective allocation and utilization of health resources. On the one hand, an aging population and lower mortality rates for children under 5-years have contributed to human life expectancy generally increasing. On the other hand, globalization and urbanization promote unhealthy lifestyles (such as tobacco and alcohol, unhealthy diet, and physical inactivity), and changes in exposure risk factors (such as indoor and outdoor air pollution), have led to chronic NCDs becoming major public health problems. What is more, new infectious diseases and the recurrence of existing infectious diseases have caused serious public health problems [76]. Therefore, achieving universal health coverage (UHC) has become the second research hotspot.

The third direction is public health education and the health workforce. Sorensen K (2015) [62] indicated that inadequate HL represents an important challenge for health policies and practices across Europe. Government subsidies play an important role in developing public health services to achieve UHC, especially in low- and middle-income countries or for internal migrants [71]. The growing popularity of the digital age has promoted the development of social media, which offered many opportunities to improve public health literacy but also created some challenges for public health [68]. We should particularly recognize the importance of an adequate and accessible health workforce to provide an integrated and effective health system and care. Jogerst K (2015) [21] put forward a list of inter-profession global health capabilities to guide extensive education programs.

## 4. Discussion

HE constitutes an increasing element in global health promotion, and more attention has been paid to relevant research. From the visualization of the domain of HE from 2011 to 2021, our study found the core articles, and the largest cluster is STH, laying emphasis on strategies of schistosomiasis control. Additionally, the two clusters labeled as global health training and community health workers were also discussed. The goals are to cultivate comprehensive, high-quality health workforces to promote professional development and improve global health. Meanwhile, the key references with the most citation bursts onward suggested that the COVID-19 pandemic has had a dramatic impact on health systems and the well-being of people and communities around the world. The topic of core health literacy for COVID-19 prevention and control has attracted widespread attention. Specifically, the topic includes several subtopics: assessment of the KAP of the public, COVID-19 epidemiology, modes of transmission and reproduction intervals, and prevention and control strategies. COVID-19 vaccine acceptance, COVID-19 psychology-related problems, the post-COVID-19 pandemic era and HE, and global emergency system management are also widely discussed.

As known, inequality in health is undesirable because it has significant effects on other parts of society, such as infectious diseases, adolescent-related health issues, alcohol or drug abuse, and violence or crime [77]. The research promotes trans-department and international government cooperation calling for action on primary health care [66,72,74], global health security [78,79], immunization vaccination [80,81], sexual and reproductive health [82,83,84,85,86,87], maternal [64,88,89], adolescent [90,91,92,93,94], and family planning [89], NCDs [95,96,97], NTDs prevention [98,99], cancer screening and treatment [100,101], etc. This would have a positive lifelong effect on health. Meanwhile, it is crucial to provide disease-matching HE for patients with lower education levels to enhance their HL, which is also one way to fight specific diseases and illnesses. In addition, using the burst detection method, we found that studies on achieving UHC and providing adequate and accessible health workforces might be more and more popular in the next few years. Specifically, as the world’s population ages rapidly, the global burden of disease (GBD) is increasing. Meanwhile, during rapid demographic and epidemiological pandemics, infectious diseases, behavioral risks, and environmental concerns could affect and threaten our health. Hence, having the ability to offer the availability of quality integrated care services and making sure adequate healthcare workers deliver effective health services is an essential element to achieving UHC [102].

Research has emphasized that increasing individual health awareness is very important for successful HE and prevention programs. The literature also advises people about healthy behaviors and lifestyle habits to prevent diseases. Healthcare organizations also need to ensure that preventive practices information is complete and adequate.

## 5. Conclusions

This study demonstrated a quantitative scientometric method and explored the progress of HE studies by using keywords and references published in this field. The results will be helpful for professional workers to understand visually the recognition modes and trends.

We analyzed a total of 10,273 eligible articles, which showed that BMC Public Health published the most prolific number of journals. The author Marco Pahor was highlighted as the most representative author in the field of HE. The University of Sydney published the most studies about HE. The USA plays an important role, creating the largest number of these studies. Specifically, the visualization shows several hotspots: disease prevalence surveys and a specific population of knowledge, attitude and practice surveys, health intervention, chronic and non-communicable management, youth-health action, sexual and reproductive health, and physical activity promotion. Furthermore, document co-citation analysis indicated that there are 10 main clusters, representing the research fronts in HE. Meanwhile, from the citations detected, COVID-19, achieving universal health coverage, public HE, and an adequate health workforce might be more popular studies in the coming years.

There are certain deficiencies in the present research. Firstly, the screened articles were merely from the Web of Science (Core Collection). Furthermore, this present study merely selected English articles, and the status of research on HE in other languages was not identified. However, we believe the results presented by the visualization by CiteSpace that the knowledge of HE is progressively arousing mounting interest. The presented research facilitates an extensive understanding of the fundamental ideas and terms that are pivotal for the developmental processes for HE and offers experts the opportunity to visualize the identification modes and tendencies.

## Figures and Tables

**Figure 1 ijerph-19-06440-f001:**
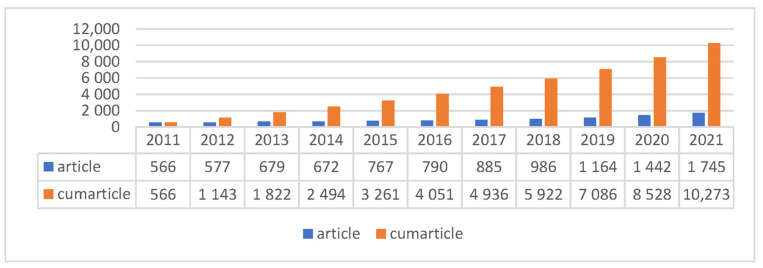
Annual publications concerning Health Education (HE) in the Web of Science (WOS^TM^).

**Figure 2 ijerph-19-06440-f002:**
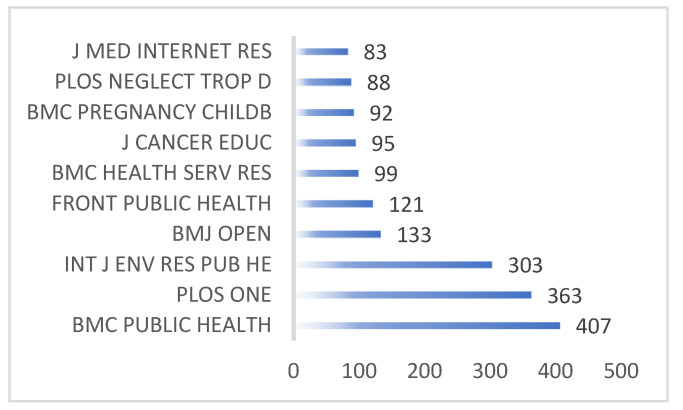
The top ten most fruitful journals.

**Figure 3 ijerph-19-06440-f003:**
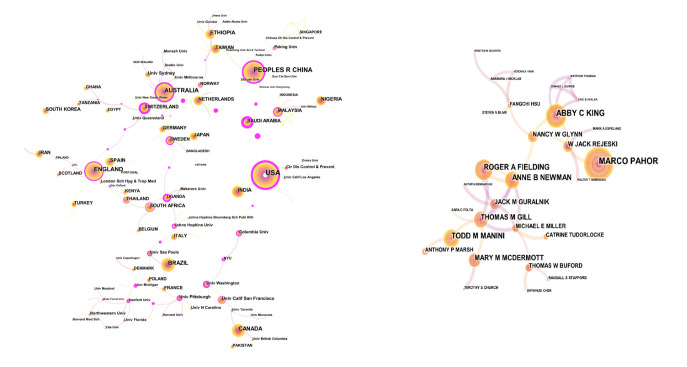
Co-authorship of HE research. The most representative author, co-institute, and country are Marco Pahor, Univ Sydney, and the USA. The size of the circle represents the number of studies published by the author, institutes, and countries. The shorter the distance between two circles, the more cooperation between the two authors. Purple rings indicate that these institutes have greater centrality (no less than 0.1).

**Figure 4 ijerph-19-06440-f004:**
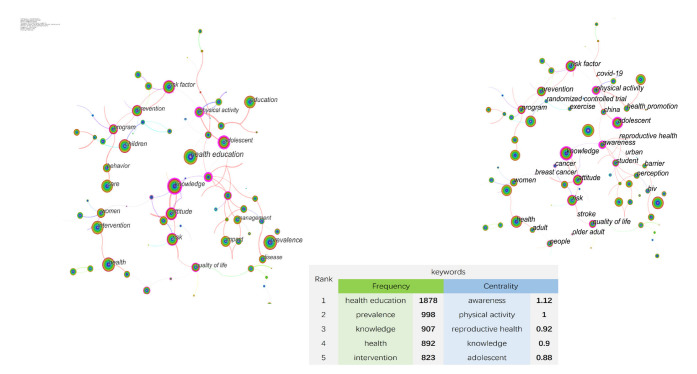
Co-occurring keywords analysis. Note: Left-frequency ranking. Right-centrality ranking. Nodes represent keywords, and the color of the lines that appear together between keywords indicate chronological order: the cold colors represent an earlier publishing time, and the warm colors mean the most recent.

**Figure 5 ijerph-19-06440-f005:**
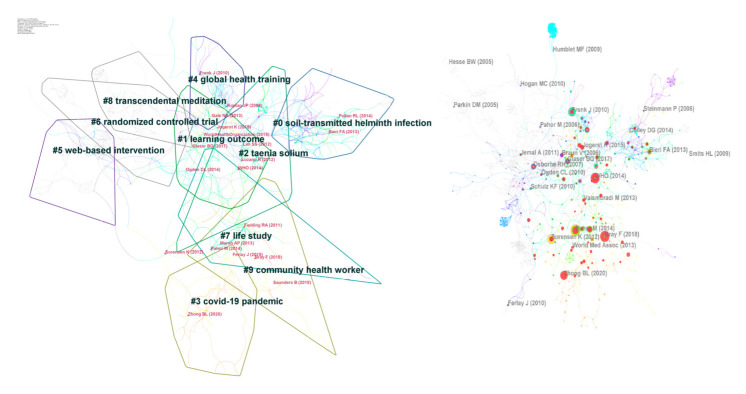
Document co-citation analysis in HE research [9,10,11,12,13,14,15,16,17,18,19,20,21,22,23,24,25,26,27,28,29,30,31,32,33,34,35,36,37,38,39,40,41,42,43,44].

**Table 1 ijerph-19-06440-t001:** Glossary of Terms.

WHO	World Health Organization
HE	health education
HL	health literacy
BC	betweenness centrality
KAP	knowledge, attitudes, and practices
NTDs	neglected tropical diseases
STH	soil-transmitted helminth infection
YLDs	years of healthy life lost
GBD	global burden of disease
NCDs	non-communicable diseases
UHC	universal health coverage

**Table 2 ijerph-19-06440-t002:** Search strategies.

Type	Search Query and Results	Results
1	(“Health Education” OR “Education, Health” OR “Community Health Education” OR “Education, Community Health” OR “Health Education, Community”) (Topic)	16,134
2	# 1 and Article (Document Type)	12,182
3	# 2 and English (Language)	11,682
4	# 3 and 2011–2021 (Year Published)	10,273

**Table 3 ijerph-19-06440-t003:** The five top-cited articles in HE.

Frequency	Author	Title	Source
52	Pahor M (2014) [9]	Effect of structured physical activity on prevention of major mobility disabilities in older adults: the LIFE study randomized clinical trial	JAMA-J AM MED ASSOC
49	Bray F (2018) [45]	Global Cancer Statistics 2018: GLOBOCAN Estimates of Incidence and Mortality Worldwide for 36 Cancers in 185 Countries	CA-CANCER J CLIN
46	WHO (2014) [10]	Global status report on violence prevention 2014	World Health Organization
41	Sorensen K (2012) [11]	Health literacy and public health: A systematic review and integration of definitions and models	BMC PUBLIC HEALTH
41	Zhong BL (2020) [12]	Knowledge, attitudes, and practices towards COVID-19 among Chinese residents during the rapid rise period of the COVID-19 outbreak: a quick online cross-sectional survey	INT J BIOL SCI

**Table 4 ijerph-19-06440-t004:** The largest 10 clusters of HE document co-citation.

Cluster	Size	Silhouette	Mean (Year)	Label (LSI)	Label (LLR)	Label (MI)
0	77	0.932	2011	risk factors	soil-transmitted helminth infection	new vision
1	66	0.773	2013	women	learning outcome	smear-positive pulmonary tuberculosis
2	62	0.831	2013	China	taenia solium	epidemiological characteristics
3	51	0.942	2019	cross-sectional study	COVID-19 pandemic	income-poor household
4	49	0.949	2009	global health education	global health training	educational engagement-a tale
5	48	0.834	2008	toddlers	web-based intervention	urban residential women
6	48	0.884	2007	controlled trial	randomized controlled trial	clinical outcome
7	48	0.935	2010	adults	life study	sleep-wake behavior
8	46	0.881	2007	effectiveness	transcendental meditation	high-risk patient
9	44	0.931	2016	impact	community health worker	southwest China

Note: there are three main methods of extracting noun phrases from the titles of articles citing clustering—term frequency by inverted document frequency (LSI), log-likelihood ratio (LLR), and mutual information (MI).

**Table 5 ijerph-19-06440-t005:** The top three articles in terms of BC.

Centrality	Author	Title	Source
0.25	Murray CJL (2012) [56]	Disability-adjusted life years (DALYs) for 291 diseases and injuries in 21 regions, 1990–2010: a systematic analysis for the Global Burden of Disease Study 2010	LANCET
0.17	Sorensen K (2012) [11]	Health literacy and public health: a systematic review and integration of definitions and models	BMC PUBLIC HEALTH
0.13	Braun V (2006) [20]	Using thematic analysis in psychology	QUAL RES PSYCHOL

The top three articles with the highest BC are Murray CJL (2012), Sorensen K (2012), and Braun V (2006), building a connection in two or more clusters.

**Table 6 ijerph-19-06440-t006:** References with recent citation bursts.

References	Citation Burst				
	Year	Strength	Begin	End	Duration (2011–2021)
Bray F (2018) [45]. Global cancer statistics 2018: GLOBOCAN estimates of incidence and mortality worldwide for 36 cancers in 185 countries. CA-CANCER J CLIN	2018	20.41	2020	2021	▂▂▂▂▂▂▂ ▂▂ ▃▃▃
Zhong BL (2020) [12]. Knowledge, attitudes, and practices towards COVID-19 among Chinese residents during the rapid rise period of the COVID-19 outbreak: a quick online cross-sectional survey. INT J BIOL SCI	2020	19.39	2020	2021	▂▂▂▂▂▂▂▂▂ ▃▃▃
Al-Hanawi MK (2020) [58]. Knowledge, Attitude, and Practice Toward COVID-19 Among the Public in the Kingdom of Saudi Arabia: A Cross-Sectional Study. FRONT PUBLIC HEALTH	2020	8.46	2020	2021	▂▂▂▂▂▂▂▂▂ ▃▃▃
Patton GC (2016) [52]. Our future: a Lancet commission on adolescent health and wellbeing. LANCET	2016	7.86	2020	2021	▂▂▂▂▂ ▂▂▂▂ ▃▃▃
WHO (2014) [10]. GLOBAL STATUS REPORT ON VIOLENCE PREVENTION 2014. World Health Organization	2014	7.06	2017	2021	▂▂▂ ▂▂▂ ▃▃▃▃▃▃
Abdelhafiz AS (2020) [59]. Knowledge, Perceptions, and Attitude of Egyptians Towards the Novel Coronavirus Disease (COVID-19). J COMMUN HEALTH	2020	7.04	2020	2021	▂▂▂▂▂▂▂▂▂ ▃▃▃
Azlan AA (2020) [60]. Public knowledge, attitudes, and practices towards COVID-19: A cross-sectional study in Malaysia. PLOS ONE	2020	7.04	2020	2021	▂▂▂▂▂▂▂▂▂ ▃▃▃
Huang CL (2020) [61]. Clinical features of patients infected with 2019 novel coronavirus in Wuhan, China. LANCET	2020	6.57	2020	2021	▂▂▂▂▂▂▂▂▂ ▃▃▃
Sorensen K (2015) [62]. Health literacy in Europe: comparative results of the European health literacy survey (HLS-EU). EUR J PUBLIC HEALTH	2015	6.37	2020	2021	▂▂▂▂ ▂▂▂▂▂ ▃▃▃
Zhu N (2020) [63]. A Novel Coronavirus from Patients with Pneumonia in China, 2019. NEW ENGL J MED	2016	5.76	2018	2021	▂▂▂▂▂ ▂▂ ▃▃▃▃▃
Victora CG (2016) [64]. Breastfeeding in the 21st century: epidemiology, mechanisms, and lifelong effect. LANCET	2015	5.76	2019	2021	▂▂▂▂ ▂▂▂▂ ▃▃▃▃
Wolf MS (2020) [65]. Awareness, Attitudes, and Actions Related to COVID-19 Among Adults With Chronic Conditions at the Onset of the U.S. Outbreak. ANN INTERN MED	2019	5.63	2020	2021	▂▂▂▂▂▂▂▂ ▂ ▃▃▃
Peres MA (2019) [66]. Oral diseases: a global public health challenge. LANCET	2014	5.63	2020	2021	▂▂▂ ▂▂▂▂▂▂ ▃▃▃
Jogerst K (2015) [21]. Identifying Interprofessional Global Health Competencies for 21st-Century Health Professionals. ANN GLOB HEALTH	2015	5.12	2018	2021	▂▂▂▂ ▂▂▂ ▃▃▃▃▃
United Nations (2015) [67]. Transforming our world: the 2030 Agenda for Sustainable Development. TRANSF OUR WORLD 203	2017	4.93	2019	2021	▂▂▂▂▂▂ ▂▂ ▃▃▃▃
Zhang XT (2017) [68]. How the public uses social media WeChat to obtain health information in China: a survey study. BMC MED INFORM DECIS	2020	4.69	2020	2021	▂▂▂▂▂▂▂▂▂ ▃▃▃
Bai Y (2020) [69]. Presumed Asymptomatic Carrier Transmission of COVID-19. JAMA-J AM MED ASSOC	2020	4.69	2020	2021	▂▂▂▂▂▂▂▂▂ ▃▃▃
Sohrabi C (2020) [70]. World Health Organization declares global emergency: A review of the 2019 novel coronavirus (COVID-19). INT J SURG	2017	4.22	2020	2021	▂▂▂▂▂▂ ▂▂▂ ▃▃▃
Zhang JY (2017) [71]. Public Health Services Utilization and Its Determinants among Internal Migrants in China: Evidence from a Nationally Representative Survey. INT J ENV RES PUB HE	2018	3.70	2019	2021	▂▂▂▂▂▂▂ ▂ ▃▃▃▃
Wang ZW (2018) [72]. Status of Hypertension in China Results From the China Hypertension Survey, 2012–2015. CIRCULATION	2013	3.67	2019	2021	▂▂ ▂▂▂▂▂▂ ▃▃▃▃
Say L (2014) [73]. Global causes of maternal death: a WHO systematic analysis. LANCET GLOB HEALTH	2017	3.23	2020	2021	▂▂▂▂▂▂ ▂▂▂ ▃▃▃
Kassebaum NJ (2017) [74]. Global, Regional, and National Prevalence, Incidence, and Disability-Adjusted Life Years for Oral Conditions for 195 Countries, 1990–2015: A Systematic Analysis for the Global Burden of Diseases, Injuries, and Risk Factors. J DENT RES	2016	2.77	2020	2021	▂▂▂▂▂ ▂▂▂▂ ▃▃▃
Chen WQ (2016) [75]. cancer statistics in China, 2015. CA-CANCER J CLIN	2018	20.41	2020	2021	▂▂▂▂▂▂▂ ▂▂ ▃▃▃

The green line means the year from 2011 to 2021, and the red line (▃) means the years of the period of burst, with one red equal to one year. For example, Chen WQ (2016) [75] has a 3-year of burst from 2011 to 2021.

## Data Availability

Not applicable.

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
