# Peer review of "Visualizing the Knowledge Domain in Health Education: A Scientometric Analysis Based on CiteSpace"

_ijerph, 2022, doi:10.3390/ijerph19116440_

Round 1
Reviewer 1 Report
Dear authors,
I congratulate you for the article entitled "Visualizing the knowledge domain in health education: a scientometric analysis based on CiteSpace" that I have had the pleasure of reviewing.
Regarding the review:
1) The work is not an article but a review.
2) The abstract must reflect which part is developed for the introduction, methodology, results and discussion section.
3) The bibliographic citation does not meet the standards of the journal.
4) Methodology: it is important to clarify whether the work is an article or a review. And based on this clarification, redo this section again.
5) Results: the figures appear pixelated. Check content and grammatical errors.
6) The discussion section must distinguish between the data found in the results and other data from scientific sources. It does not conform to the standard of the magazine.
I request modifications.
Reviewer 2 Report
This manuscript reviews scientific literature and presents research trends in health education. Some scentometric analyses have been conducted to visualize and examine the selected articles. This methodology has been widely used in bibliometrics. The manuscript shows some potential but this manuscript needs to be strengthened in addressing several items.
- Scientometrics has been intensively used in a variety of domains. I suggest that authors review these articles listed below to have a better understanding of scientometrics to improve the literature review section. Basic concepts, different scientometric methods and tools, and results are introduced and discussed in these papers.
-
- Ding, W., & Chen, C. 2014. Dynamic topic detection and tracking: A comparison of HDP, C‐word, and cocitation methods. Journal of the Association for Information Science and Technology, 65(10), 2084-2097.
- Small, H. 1973. Co-citation in the scientific literature: A new measure of the relationship between two documents. Journal of the American Society for Information Science, 24:265–69.
- Small, H. 1980. Co-citation context analysis and the structure of paradigms. Journal of Documentation, 36 (3): 183–96.
- Small, H., & Griffith, B.C. 1974. The structure of scientific literatures: I. Identifying and graphing specialties. Science Studies, 4:339–65.
- White, H. D., and K. W. McCain. 1998. Visualizing a discipline: An author co-citation analysis of information science, 1972–1995. Journal of the American Society for Information Science, 49 (4): 327–55.
- I don’t see much introduction/background in the manuscript. There are already many published academic papers discussing health education or similar topics using scientometric methods. I strongly suggest that it’s necessary to have a thorough or at least decent introduction and literature review on this.
- The structure and content in results section is somehow confusing and problematic.
- Section 3.3 is co-occurring keywords analysis but the content in this section is about clusters generated in CiteSpace.
- Section 3.4 is document co-citation analysis but the majority part in this section is about clusters generated and articles with high betweenness centrality. Also, it should be table 3 not table 9 on page 6.
- One of the essential components in CiteSpace is cluster analysis. I encourage authors to expand this part with more details and thorough discussion on identified interesting topics. What are new topics and what are old ones? What findings/insights can be captured from the clustering results? I also suggest that authors can play with the CiteSpace parameters to have more meaningful cluster labels generated if possible. A figure with cluster view is missing here. It’d be very helpful to illustrate the clustering results. Several metrics are provided in the cluster identification process. Some detailed information would be useful. For example, what metric did authors use and why? Or use all the metrics and compare the labels generated from metrics in each cluster. Make this part as an individual section in results.
- Betweenness centrality should be considered as an individual sub-section as well in results. More importantly, this particular network measure can identify the important nodes that connect different groups of nodes or clusters. That said, the summary of the articles with high betweenness centrality is not enough. It’d be important to have a discussion of what different clusters or groups of articles a high betweenness centrality article connects. A discussion with possible hidden connection would make the manuscript appealing.
- Table 5 is unnecessarily long. It’d be enough to present top 5, top 10, or selected references. Other references can be presented in appendix if authors think it’s necessary.
- All CiteSpace figures need a higher resolution.
- I suggest that authors carefully review the manuscript and make sure no grammatical errors.
Reviewer 3 Report
The present manuscript explores an important and timely topic in health education that will be of interest to a wide range of audience. Below I have a couple of comments I hope the authors could address before I can recommend the manuscript for publication:
- It is unclear what the hypotheses were. Could the authors include some text on their specific hypotheses/predictions in the abstract and introduction?
- Were statistical analyses performed? If so, please elaborate on this point in the method section. If not, please explicit discuss why this is the case given the current questions being explored in the manuscript.
- Please include more descriptions to each of the figure caption to indicate the main take-away message of each figure.
- Please consider enlarging the text/legend within each figure especially figure 2 and figure 3 as they are currently difficult to read.
Round 2
Reviewer 1 Report
1. The study presents the results of original research.
2. Results reported have not been published elsewhere.
3. Experiments, statistics, and other analyses are performed to a high technical standard and are described in sufficient detail.
Reviewer 3 Report
The authors have carefully and thoroughly responded to all of my previous comments.